# Deep Learning-Based Electric Field Enhancement Imaging Method for Brain Stroke

**DOI:** 10.3390/s24206634

**Published:** 2024-10-15

**Authors:** Tong Zuo, Lihui Jiang, Yuhan Cheng, Xiaolong Yu, Xiaohui Tao, Yan Zhang, Rui Cao

**Affiliations:** Key Laboratory of Aperture Array and Space Application, East China Research Institute of Electronic Engineering, Hefei 230088, China; emanonxmu@sina.com (L.J.);

**Keywords:** microwave tomography, stroke detection, convolutional neural network, degree of freedom, Born Iterative Method

## Abstract

In clinical settings, computed tomography (CT), magnetic resonance imaging (MRI), or positron emission tomography (PET) are commonly employed in brain imaging to assist clinicians in determining the type of stroke in patients. However, these modalities are associated with potential hazards or limitations. In contrast, microwave imaging emerges as a promising technique, offering advantages such as non-ionizing radiation, low cost, lightweight, and portability. The primary challenges faced by microwave tomography include the severe ill-posedness of the electromagnetic inverse scattering problem and the time-consuming nature and unsatisfactory resolution of iterative quantitative algorithms. This paper proposes a learning electric field enhancement imaging method (LEFEIM) to achieve quantitative brain imaging based on a microwave tomography system. LEFEIM comprises two cascaded networks. The first, based on a convolutional neural network, utilizes the electric field from the receiving antenna to predict the electric field distribution within the imaging domain. The second network employs the electric field distribution as input to learn the dielectric constant distribution, thereby realizing quantitative brain imaging. Compared to the Born Iterative Method (BIM), LEFEIM significantly improves imaging time, while enhancing imaging quality and goodness-of-fit to a certain extent. Simultaneously, LEFEIM exhibits anti-noise capabilities.

## 1. Introduction

Stroke is an acute cerebrovascular disease that causes an interruption of blood supply to critical areas of the brain, resulting in a lack of oxygen and nutrients in brain tissue. Data show that stroke is gradually becoming the leading cause of death both in China and worldwide, as well as a leading cause of disability among Chinese adults. Stroke can be divided into ischemic stroke and hemorrhagic stroke. Rapidly identifying the type of stroke and correctly using thrombolytic drugs are the key to treatment; otherwise, irreversible damage will occur. At present, CT, MRI, and PET are the major diagnosis methods used in brain imaging. These methods require the use of large equipment, which is costly and time-consuming. In addition, CT produces ionizing radiation, and the use of contrast agents may be toxic to the kidneys. PET requires the injection of radioactive substances in advance, which poses a potential threat to the human body. In recent years, electromagnetic tomography (EMT) using microwave signals, also known as microwave tomography (MWT), has emerged as a promising brain imaging modality. It is not only a cost-effective, portable, and fast imaging solution, but also offers non-ionizing and non-invasive diagnosis.

In 2008, Serguei Y. Semenov et al. studied the feasibility of using microwave tomography brain imaging to detect stroke [1]. They proposed a method using a 2D model of the human head to represent the major tissues of the head (including skin, skull, cerebrospinal fluid, gray matter, white matter, and stroke regions) using a nonlinear Newtonian reconstruction method. The results show that multi-frequency MWT has the potential to significantly improve imaging results. In the same year, Hana Trefn´a et al. designed an antenna array consisting of triangular microstrip antenna elements, which contained a V-shaped groove and short-circuit walls for stroke detection [2]. The simulation results showed that there was a 3 dB difference between stroke and normal conditions, which could provide sensitivity for detecting changes. In 2009, Colin Gilmore et al. proposed FD-CSI based on CSI and verified the algorithm using a simple brain model [3]. In 2010, Amer Zakaria et al. combined FEM with CSI to enhance microwave tomography of conductive shells [4]. In 2013, N. Priyadarshini et al. designed a brain imaging model using COMSOL at 1 GHz, using CSI as a quantitative reconstruction [5]. In the same year, Malyhe Jalilvand et al. analyzed a five-layer brain structure based on ultra wide band radar and derived the design requirements for a UWB radar system for head imaging [6]. David Ireland et al. conducted simulation experiments using BIM, achieved imaging at 850 MHz, and tested its anti-noise capability [7]. In 2014, M. Jalilvand et al. used an iterative algorithm based on Gauss–Newton for brain reconstruction. The initial guess started from the background of a homogeneous medium and iterated 10 times, converging to the actual value [8]. Serguei Semenov et al. developed BRIM-G1, a five-layer imaging system consisting of 160 ceramic-loaded waveguide antennas, using a nonlinear Newton iteration scheme for imaging [9]. In 2016, İsmail Dilman et al. used CSI for imaging at 500 MHz, 800 MHz, and 1000 MHz based on simulation and successfully determined a square bleeding area with a side length of 2.6 cm [10]. These researchers also completed work on quantitative microwave imaging of the brain [11,12,13,14,15]. In conclusion, commonly used imaging algorithms include Contrast Source Inversion (CSI) [16], the Born Iterative Method (BIM) [17], and the Subspace-Based Optimization Method (SOM) [18], etc.

In recent years, with increases in computing power and the development of deep learning (DL), microwave tomography of the brain using deep learning is set to take center stage. The application of deep learning methods for brain imaging is mainly categorized into two categories. The first is based on microwave systems. In 2020, researchers proposed a method that uses an antenna array system to capture microwave signals and learn normal versus abnormal brain scattering signals through an unsupervised machine learning model, learning the difference between normal and abnormal signals [19]. In 2021, Ahmed Al-Saffar’s [20] research team proposed a complex-valued model that implements the conversion of simulated data to experimental data and then reconstructs the dielectric constant distribution map based on this. Another category is the use of deep learning methods as image segmentation for clinical tools such as in CT or MRI [21,22]. Since these methods are not based on microwave signals, they cannot avoid the inherent defects of microwave imaging, nor take advantage of its advantages. In addition, some other studies that combine deep learning with electromagnetic backscatter have also achieved certain results in simulation. Costanzo, S [23] used UNet for brain cancer imaging and enhanced the unsatisfactory BIM imaging results to achieve coherent imaging. Yao, H. M. [24] designed a Fast Electromagnetic Inverse Solver using a conditional deep convolutional generic aggressive network. However, there are still many questions in this broad field [25,26,27].

In general, the application of deep learning in this field is still quite rare. On the one hand, due to the lack of clinical data, deep learning is not suitable for use; on the other hand, due to the fact that deep learning has certain black-box attributes and weak interpretability, it is difficult to meet the high accuracy and reliability required by the medical field. Given the drawbacks stated above, this paper proposes a method combining experiment and simulation to construct a cascade network structure to realize dielectric constant reconstruction. The first network uses a mixture of experimental and simulation data for learning to enhance the generalization performance of the model. The second network learns the mapping relationship between the scattered field and the dielectric constant distribution, and understands the physical principles through the learning ability of the model.

The structure of this paper is as follows: Section 2 introduces the problem equation and then describes the proposed LEFEIM and its evaluation scheme. Section 3 describes the experimental design, including the imaging system, head phantom production, and numerical simulation. Section 4 demonstrates the performance of LEFEIM and analyzes the results in comparison with the traditional method BIM. A conclusion follows in Section 5.

## 2. Learning Electric Field Enhancement Imaging Method

In response to the problems described in the previous section, this paper proposes LEFEIM to improve the performance of brain microwave tomography. This section describes the method from three perspectives, including, firstly, an analysis of the physical basis of the electromagnetic problem, and then the introduction of the structure of LEFEIM, and lastly, an assessment methodology for the proposed LEFEIM method.

### 2.1. Equation of the Problem

Figure 1 shows a schematic diagram of microwave tomography. Under the incoming electromagnetic waves, the motion state of unknown object particles changes, and this process is called scattering. The essence of microwave tomography is to calculate the dielectric constant distribution of the domain of interest (DOI) through changes in the electromagnetic field.

Here, we consider a two-dimensional imaging scenario, where the incident wave is a transverse magnetic (TM) wave, and the scattered field satisfies the scalar Helmholtz equation. Assuming a uniform dielectric background with the dielectric constant $\epsilon_{0}$ and magnetic permeability $\mu_{0}$, a scatterer with relative permittivity $\epsilon_{r}\left(r\right)$ (the brain in this study) locates at the center of the imaging domain *D*. Electromagnetic waves illuminate objects from different angles. After the incident wave interacts with the scatterer in the imaging domain, it is received by the receiving antenna located at $r_{i}$ with i=1,2,…,N. Here, *N* denotes the total number of receivers for each incidence. To describe and calculate this scattering process, we introduce the Lippmann–Schwinger equation:(1)$$E^{tot}\left(r\right)=E^{inc}\left(r\right)+i\omega \mu_{0}\int_{D}G\left(r,r^{\prime }\right)\left(-i\omega \epsilon_{0}\left(\epsilon_{r}\left(r^{\prime }\right)-1\right)E^{tot}\left(r^{\prime }\right)\right)dr^{\prime }\;r\in D$$

This equation describes the wave-scatterer interaction, where $E^{tot}\left(r\right)$ represents the total electric field, $E^{inc}\left(r\right)$ is the incident field, ω is the angular frequency, $G(r,r^{\prime })$ is the Green’s function, which can be rewritten as a zero-order first-type Hankel function $\frac{i}{4}H_{0}^{\left(1\right)}(k_{0}|r-r^{\prime }\left|\right)$ under the two-dimensional scenario, and $\epsilon_{r}\left(r^{\prime }\right)$ is the relative permittivity of the object. The right-hand side of the equation can be divided into two main parts. The first part describes the free wave function when there is no electric field acting on it, i.e., the state prior to the action of the electric field. The second part, denoted as $E^{sca}\left(r\right)$, describes the scattering field of interactions within the scattering region in the presence of scatterers, which can be written as:(2)$$E^{sca}\left(r\right)=i\omega \mu_{0}\int_{S}G\left(r,r^{\prime }\right)\left(-i\omega \epsilon_{0}\left(\epsilon_{r}\left(r^{\prime }\right)-1\right)E^{tot}\left(r^{\prime }\right)\right)dr^{\prime }\;r\in S$$

The physical meaning of scattering field is the electric field generated by an induced current, due to the existence of the scatterer. $(-i\omega \epsilon_{0}\left(\epsilon_{r}\left(r^{\prime }\right)-1\right)E^{tot}$ represents the induced current density and $\left(\epsilon_{r}\left(r^{\prime }\right)-1\right)$ denotes the contrast χ. The contrast χ is the unknown to be solved for reconstruction and also the key factor in the generation of scattering fields. Previous studies have proven that solving contrast through the above equation is unique and stable [28]. However, the total amount of available data, that is, the product of the number of transmitters and receivers, is less than the number of unknowns. Meanwhile, the number of essentially independent equations (i.e., the product of the number of transmitters and the degrees of freedom of the scattering operator) is far less than the degrees of freedom of the unknowns. This study concentrates on the two dimensional scenario, and the degrees of freedom (DoF) of the observables are approximated as [29]:(3)DoF≈2Kα
where *K* denotes the wave number, and α denotes the radius of the circular antenna array. The degree of freedom represents the minimum number of observables required for imaging reconstruction. The degrees of freedom of the unknowns are much larger than the degrees of freedom of the observables, which leads to the inverse scattering problem with serious nonlinearity and ill-posedness. Therefore, when considering using DL for dielectric constant reconstruction, a limited number of observations are used as inputs for the model. The first part of LEFEIM is designed to increase the degrees of freedom of the observables to realize the effect of electric field enhancement. The enhanced electric field is then used as an input to the dielectric constant reconstruction. Meanwhile, in the second part of the model, in order to reduce the time spent on reconstructing the dielectric constant, a deep learning architecture is used for fast computation.

### 2.2. LEFEIM

In this subsection, based on the previous analysis, a LEFEIM is designed to realize dielectric constant reconstruction. The first network aims to achieve electric field enhancement by increasing the DoF of the data. A convolutional neural network (CNN) with an added self-attention module is constructed as the first network of LEFEIM. The detailed structure is as shown in Figure 2.

A four-layer CNN was constructed with kernel size set to 3, padding set to 1, and the activation function using RELU, and finally the model was optimized by back-propagation. Stacking multiple 3 × 3 small convolution kernels can have the same receptive field as large convolution kernels, but the parameters and computational complexity of small convolution kernels are less. Stacking multiple 3 × 3 small convolution kernels can introduce more nonlinearity compared to large convolution kernels, resulting in better reconstruction of the scattering field. A self-attention module is added to the second layer to give the model a reconstruction of the electric field that can be more focused on the brain region. The last layer of this network is a fully connected layer to increase the number of input channels and enhance feature recognition, and to uplift the input one-dimensional signal to two dimensions. The size of the input layer is n × the number of receiving antennas, where n represents the number of antenna positions. The output layer is the enhanced electric field map.

According to the Lippmann–Schwinger equation, the distribution of the dielectric constant in the region of interest directly affects the scattering process in the presence of an external electric field. Although the incident field does not directly affect the latter half of the scattering field, indirect effects can also occur due to subsequent interactions. Therefore, the solution of the dielectric constant distribution is solved as an implicit inverse problem, which needs to consider the effects of both the incident field and the total field, i.e., the combined effects of the free wave function and the external electric field. Therefore, in order to realize an accurate reconstruction of the dielectric constant distribution, a network structure is designed, as shown in Figure 3. The network structure is a U-Net with an added attention gate [30]. The classical U-Net is a method used for segmentation of biomedical images as a classifier for pixel-level problems. U-Net gets its name from its U-shaped network structure, where the left side consists of a continuous convolutional layer with a pooling layer, a step also known as encoder, and the right side consists of successive up-sampling with a convolutional layer and a feature fusion step called decoder. An attention gate is added to the network architecture for implicitly capturing key information. This means that there is no need to provide additional information about brain location or other supplementary means to achieve the effect of suppressing irrelevant areas and enhancing the reconstruction of regions of interest. The learning objective of the network is the mapping between the two-channel field (scattered field, incident field) and the contrast, which is essentially an inverse problem solver to improve the traditional algorithms’ pain points such as high computational complexity, its time-consuming nature, and the poor results attained through the deep learning model.

The signals captured by receiving antennas are input while the model is learning. The signals are fed into the first network of the model and the electric field distribution of the scattered field is output by the trained network with a resolution of 400 × 400 (mm). Then, the scattered field and incident field information output from the first network of LEFEIM is used as the input to the second part of the model, and the dielectric constant distribution can be calculated from the input two-channel electric field information. The two networks are cascaded to form LEFEIM.

The main parameter settings considered include the learning rate, the batch size, and the number of iterations. The adjustment of the learning rate directly controls the order of magnitude of the network gradient update during the training process, which directly affects the training speed and convergence performance of the model. This is one of the most important hyper-parameters, and the learning rate is set to 0.005. The batch size refers to the number of samples used in one iteration, and a smaller batch size can improve the convergence speed of the model, but may increase the amount of computation and memory consumption required. On the other hand, a larger batch size can reduce the use of computational resources and memory consumption, but may affect the training speed and convergence performance of the model. Considering the limited memory of the server, the selected batch size was 3. The number of iterations refers to the number of times the model is trained. The higher the number of iterations, the better the training effect of the model. According to the loss function, the number of iterations was set to 50. The optimizer is a method to find the minimum value of the loss function, and this study selected the Adaptive Moment Estimation (Adam) optimizer. It has the advantages of fast convergence speed and good learning effect. It should be noted that the Adam will adaptively modify the learning rate of different parameters during the training process, so the learning rate setting refers to the initial learning rate.

### 2.3. Evaluation Metrics of LEFEIM

The LEFEIM proposed in the previous section is a cascade network consisting of two different sub-networks. The two sub-networks have different structures and learning tasks. In order to unify the quantitative evaluation criteria and the assessment index, the results of the method are used as the assessment criteria, and there is no additional assessment of the sub-networks conducted separately. The quantitative analysis is accomplished by comparing the gap between the proposed method and the traditional method through unified assessments. For the inverse scattering problem, both BIM and CSI belong to the optimization method. The principle is essentially the solution of the regression task, so the commonly used evaluation method of the regression algorithm is introduced here to carry out the quantitative analysis. In order to observe the reconstructed outliers, the root mean square error is introduced as an evaluation index. The fitness of the proposed model is judged by the coefficient of determination.

#### 2.3.1. Outlier Assessment

To quantitatively evaluate LEFEIM, two sets of evaluation methods were used. The first one is based on the root mean square error (RMSE). By calculating the RMSE of the whole imaging domain, the reconstruction performance of the electric field distribution and the dielectric constant distribution are evaluated separately. The RMSE is calculated by the equation: (4)$$RMSE=\sqrt{\frac{\sum_{i=1,j=1}^{N,M}{\left(S_{R_{i,j}}-{\bar{S}}_{R_{i,j}}\right)}^{2}}{N\times M}}$$
where *N* and *M* represent the lateral resolution and vertical resolution of the imaging domain, respectively; $S_{R_{i,j}}$ denotes the reconstructed electric field value or dielectric constant value located at (i,j); and ${\bar{S}}_{R_{i,j}}$ denotes the actual electric field value or dielectric constant value at (i,j). The root mean square error is more sensitive to larger outliers, so it can be easily analyzed whether there are more significant outliers in the imaging domain.

#### 2.3.2. Goodness of Fit Assessment

For the regional evaluation of brain models, the coefficient of determination, denoted $R^{2}$, is chosen as the metric for evaluating regression tasks. The purpose of brain imaging is to reconstruct the dielectric constant distribution, i.e., to establish the mapping of the received signal to the electric field distribution and the electric field distribution to the dielectric constant distribution, which is clearly a regression task. Thus, the coefficient of determination is used as a statistical measure of the goodness of fit of the regression model. The coefficient of determination indicates how well the regression model fits the data set, i.e., it indicates the percentage explained by the regression model. For the study in this paper, a higher coefficient of determination proves that the proposed model works better. As indicated by Equation (Equation 5), the steps for calculating the coefficient of determination are as follows: calculate the sum of squares due to error (SSE) of the residuals between the predicted electric field values or dielectric constant values and the true values.
(5)$$SSE=\sum {\left(y_{i}-{\hat{y}}_{i}\right)}^{2}$$

For the ith observation point, the difference between the true value and the estimated one is the ith residual, and the residual sum of squares is the sum of the residuals of all the observation points. The closer the residual sum of squares is to 0, the better the model fits, and the more successful the data prediction is. Then, the total sum of squares (SST) and the sum of squares due to regression (SSR) are calculated according to Equations (6) and (7).
(6)$$SST=\sum {\left(y_{i}-{\bar{y}}_{i}\right)}^{2}$$
(7)$$SSR=\sum {\left({\hat{y}}_{i}-{\bar{y}}_{i}\right)}^{2}$$

Clearly, the three items satisfy the following relationship: (8)SST=SSR+SSE

The coefficient of determination $R^{2}$ can be expressed as follows: (9)$$R^{2}=1-\frac{SSR}{SST}=\frac{SSE}{SST}=\frac{\sum_{i=1,j=1}^{N,M}{\left(y_{i,j}-{\hat{y}}_{i,j}\right)}^{2}}{\sum_{i=1,j=1}^{N,M}{\left(y_{i,j}-{\bar{y}}_{i,j}\right)}^{2}}$$

The coefficient of determination can be commonly understood as using the mean as the error benchmark to compare the relationship between the prediction error and the mean benchmark error. Generally speaking, the closer $R^{2}$ is to 1, the more the model can explain the variation in the data, and the better the model fits. When $R^{2}$ is close to 0, it means that the model cannot explain the variation in the data and the fitting effect is poorer; when $R^{2}$ is equal to 0, it means that the fitted curve cannot explain the variation in the target variable. When $R^{2}$ is equal to 1, it means that the fitted curve completely explains the variation in the target variable, and the fitting effect is very good.

## 3. Experiments with 3D-Printed Phantoms

According to the proposed LEFEIM, both the dependence with the experimentally collected S-parameters and the electric field distribution are needed at the same time. Therefore, this Section is divided into three parts. The first part describes the experimental setup and the construction of the imaging system. The second part introduces the head phantom based on ‘Zubal’ [31], which is used to determine the shape of the head phantom. The third part uses the Method of Moment (MoM) to calculate the electric field.

### 3.1. Imaging System

For the problem of stroke imaging, it is necessary to construct a system to realize the functions of information acquisition and data processing through a scientific and feasible design. Thus, a system is designed as shown in Figure 4. Following the previous section, the problem is set up as a reconstruction of the distribution of dielectric constants on a two-dimensional plane. It is assumed to be an infinite extension in the z-direction, and the reconstruction of the area outside the imaging domain boundary is not considered.

The system uses 16 Vivaldi antennas uniformly distributed on the boundary of a circular array of 0.19 m radius with a selected center frequency at 1.5 GHz. For the control of this system, the python program is first used to send the commands to the 64-way relay through the serial protocol according to a certain time interval, and then the electrical signals are transmitted to the SP16T. The selection of the transmitting antenna and the receiving antenna is completed by the SP16T, and then the results are transmitted to the Vector Network Analyzer (VAN8720ES) through the SP16T. Finally, the vector network transmits the raw data to the host computer. It should be noted that the SP16T can only select a single transmitter and a single receiver at a time, so the control signal of the relay needs to constantly switch between transmission and reception to achieve 16 transmissions and 15 receivers. The above logic enables the antennas to illuminate the head phantom at multiple angles and to acquire the received signals for reconstruction. The whole system also includes the host computer, power supply, cables, head phantom, and other devices. The picture of the whole system is shown in Figure 5. The system parameters are as shown in the following Table 1.

### 3.2. Head Phantom

The fabrication of the head phantom was a most important task in this study. In terms of previous papers, the production of head phantoms can be divided into the three main categories. The first class is the direct use of 3D printing for the entire head phantom; the material of the created phantom is set as the printing material, and the head phantom is fully 3D printed. The problem with this method is that the use of such materials for 3D printing is more demanding in terms of meeting both requirements surrounding the thickness of the 3D print and the dielectric constant, and it is not easy to modify once completed. The second class of research uses liquid materials directly as head phantoms, which allows for easy modification of the shape and is relatively affordable. The third class uses 3D printing to create the shell of the phantom and fills it with liquid dielectric mimic materials. This method reduces both cost and error. Based on these considerations, the third method was used to create the head phantom in this study.

A mold for the head phantom needed to be designed first. The shape of the mold used is referred to as a Zubal head phantom. The Zubal head phantom is a scan of a normal male’s head, which consists of a total of 256 × 256 × 128 pixels in the x, y, and z axis, respectively. The resolution in the three directions is 1.1 mm × 1.1 mm × 1.5 mm, i.e., the actual scans are 256 × 1.1 mm in the x direction, 256 × 1.1 mm in the y direction, and 128 × 1.5 mm in the z direction. According to the previous section, only the reconstruction of the dielectric constant in the two-dimensional plane was considered in this study, so slice #39 in the xoy plane was chosen, denoted as slice #39 later. Since the resolution of the original data was not very high and the edges were not very clear, some reasonable adjustments were made to slice #39, and the before and after comparison is shown in Figure 6.

The left side shows the original slice #39, and the right side shows the image after adjustment. The human brain in the slice is normal tissue, so it is necessary to place the stroke model in a reasonable position in subsequent experiments. The adjusted slice was used as a standard for subsequent production and the 3D map. The model was exported to .STL format for 3D printing. Shown Figure 7.

The mold was designed with a hollow structure, with a vertical height of 100 mm. The thinnest part was 1.46 mm and the thickest part was 3.56 mm. After the mold was 3D-printed, gelatin-based brain mimic materials were made to fill the molds. In order to ensure that the experiments were conducted in an environmentally friendly and safe manner, the study used the phantom equation according to [32].

The head phantom designed in this study had six layers of tissue, including ischemic stroke, cerebrospinal fluid, gray matter of the brain, white matter of the brain, skull, and skin. All these brain tissues were non-homogeneous substances with dispersion effects. Therefore, the relative dielectric constants of different tissues at different frequencies needed to be calculated using a specific dielectric constant model. There are many methods for calculating the dielectric properties of tissues, and the second-order Debye model was adopted in this study for reasons of compromise between computational complexity and accuracy [33]: (10)$$\epsilon_{s}\left(\omega \right)=\epsilon_{\infty }+\frac{\epsilon_{d}-\epsilon_{m}}{1+j\omega \tau_{1}}+\frac{\epsilon_{m}-\epsilon_{\infty }}{1+j\omega \tau_{2}}+j\frac{\sigma_{s}}{\omega \epsilon_{0}}$$

In the above model, the complex permittivity $\epsilon_{s}\left(\omega \right)$ is expressed as a function of frequency ω, where $\epsilon_{\infty }/\epsilon_{s}$ is the relative permittivity at infinite and zero frequencies, respectively; $\sigma_{s}$ represents the conductivity; and $\epsilon_{d},\epsilon_{m},\tau_{1},\tau_{2}$ are fitting parameters. Table 2 shows the calculated dielectric constant based on this second-order Debye model, and the parameters for calculating the dielectric constant of the organization were as follows.

After determining the shape and relative dielectric constant of the mold, the head phantom needed to be fabricated. The materials that needed to be used to fabricate phantoms representing cerebrospinal fluid, gray matter, white matter, skull, skin, and ischemic stroke tissue are shown in Table 3. It should be noted that 1 mL of surfactant as well as propanol needed to be added to the fabrication of each type of tissue to prevent air bubbles in the final product, as well as to dissolve it uniformly.See Table 3 for details.

Studies on the dielectric constant of ischemic versus hemorrhagic stroke state that the difference between the dielectric constant of the stroke region and that of the normal region is 10–25% [33]. The ratio of kerosene to safflower oil in the above table is 1:1. The process of phantom fabrication was as follows: firstly, water mixed with gelatin was heated to 80 °C, then kerosene safflower oil mixture was added, and the mix was heated to 100 °C. After complete dissolution, it was cooled down to 50 °C prior to the addition of surfactant and propanol, and then cooled down to room temperature. The finished product is shown in Figure 8.

### 3.3. Forward Simulation

The experimentally collected data contained only the S-parameters of the receiving antenna, and in order to obtain the electric field distribution in the imaging domain, it was necessary to establish a numerical artifact and compute it by the MoM [34]. In this section, a forward solver based on the setup of this experimental scenario is presented.

According to the physical meaning, we replaced $\left(\epsilon_{r}\left(r^{\prime }\right)-1\right)$ in Equations (1) and (2) with contrast $\chi (r^{\prime })$, and $i\omega \mu_{0}$ with wave number ${k_{0}}^{2}$, to obtain the following equation:(11)$$E^{tot}\left(r\right)=E^{inc}\left(r\right)+{k_{0}}^{2}\int_{\Omega }G\left(r,r^{\prime }\right)\chi \left(r^{\prime }\right)E^{tot}\left(r^{\prime }\right)dr^{\prime }$$
(12)$$E^{sca}\left(r\right)={k_{0}}^{2}\int_{\Omega }G\left(r,r^{\prime }\right)\chi \left(r^{\prime }\right)E^{tot}\left(r^{\prime }\right)dr^{\prime }$$
$\chi \left(r^{\prime }\right)E^{tot}\left(r^{\prime }\right)$ represents the equivalent current of $J(r^{\prime })$.
(13)$$J\left(r^{\prime }\right)=\chi \left(r^{\prime }\right)E^{tot}\left(r^{\prime }\right)$$

Changing Equations (1) and (2) to the above form enabled us to proceed to the next step of derivation. In general, if there is an only incident field and scatterer distribution, the analytical solution of the above equations is not obtained, so we used the method of moments to discretize the above integral equations into linear equations. The method of moments was considered by discretizing the imaging domain into an *M* small cell, and dividing the whole region into an *M* cell of the same size. It was assumed that the cells were small enough that the electric field was uniformly distributed within each cell. Then, the above equation was discretized as: (14)$$E_{n1,n2}^{tot}=E_{n1,n2}^{inc}+\sum_{n1^{\prime }=n2^{\prime }=1}^{M1,M2}G_{D;n1,n2;n1^{\prime },n2^{\prime }}J_{n1^{\prime },n2^{\prime }}$$
where n1, n2 are the indexes of the small cell, and $J_{n1^{\prime },n2^{\prime }}$ denotes the source of the induced current. $G_{D}$ is the Green’s function that maps the discrete form of the induced current. Combining the equations for each small cell, a new equation representation was obtained: (15)$${\bar{E}}^{tot}={\bar{E}}^{inc}+{\bar{G}}_{D}\cdot J$$

Similarly, ${\bar{E}}^{sca}$ can be expressed by discretization as: (16)$${\bar{E}}^{sca}={\bar{G}}_{S}\cdot J$$
where ${\bar{E}}^{tot}$ is ${\left[{\bar{E}}_{1}^{tot},{\bar{E}}_{2}^{tot},\ldots ,{\bar{E}}_{M}^{tot}\right]}^{T}$, and similarly, ${\bar{E}}^{inc}$ and ${\bar{E}}^{sca}$ are expressed as combined terms for different points. It should be noted that the first Green’s function region indicates the imaging domain, while the second Green’s function indicates the domain boundary where the receiver antennas locates. The induced current for all cells can also be expressed as: (17)$$\bar{J}={\bar{\chi }}_{diag}\cdot {\bar{E}}^{tot}$$

Combining these two equations gives: (18)$$\bar{J}={\bar{\chi }}_{diag}\cdot \left({\bar{E}}^{inc}+{\bar{G}}_{D}\cdot J\right)$$

Combining the induced currents is not difficult to derive: (19)$$\bar{J}={\left(\bar{I}-{\bar{\chi }}_{diag}\cdot {\bar{G}}_{D}\right)}^{-1}\left({\bar{\chi }}_{diag}\cdot {\bar{E}}^{inc}\right)$$
where $\bar{I}$ denotes the unit matrix. Forward calculations were performed by constructing scenarios consistent with the actual situation through simulation, and the electric field distribution was obtained using MoM.

## 4. Experimental Results and Analysis

In this section, the results are analyzed and LEFEIM is evaluated across several dimensions. These are the main sections: data calibration, quantitative analysis of imaging results, and noise immunity analysis. The following training results were obtained from an Intel(R) Core(TM) i7-11700K CPU, 32 G RAM, and an NVIDIA GeForce GTX 1650 SUPER GPU, and the training and testing of the model were performed on the GPU, while the forward simulation of the electromagnetic field and BIM were run on the CPU.

### 4.1. Data Calibration

The parameters collected by the vector network analyzer are S-parameters, and the data need to be calibrated taking into account practical requirements. Given the experimental system used in this study, the receiving antenna and the transmitting antenna were fixed in a certain position so the mechanical structure was stable and the electrical connection remained unchanged during the measurement process. Therefore, it was necessary to use the incident field as a reference, and the calibration was completed by calculating the correction factor ${CAL}_{F}$ for all positions. The calculation equation was as follows: (20)$${CAL}_{F}=E_{i,sim}/E_{i,exp}$$

The above equations ${CAL}_{F}$, $E_{i,sim}$ and $E_{i,exp}$ are 16 × 16 complex matrices. ${CAL}_{F}$ denotes the correction factor, and $E_{i,sim}$ and $E_{i,exp}$ represent the incident field data calculated using the MOM and the incident field data collected experimentally, respectively. Subsequent calibration was performed by simply multiplying the received data with the correction factor.

### 4.2. Dataset and Results

The composition of the dataset included simulation data as well as experimental data. A total of 19 scenarios were set up, including a set of normal scenarios and nine different stroke locations with 5 mm or 10 mm radii, comprising different scenarios of both hemorrhagic and ischemic stroke.

In order to increase the richness and completeness of the data while increasing the observation angle, the object was rotated in the clockwise direction at 20° intervals during the simulation, and the relative positions of the transmitting antenna and the receiving antenna were kept constant in each group. One out of every five groups in the dataset was used as a test set, and the remaining four groups were used as training sets. The number of training and testing sets were represented by 80% and 20% of the total dataset. The training time of the model was about 48 h 39 min. The following results all reflect the model’s performance on the test set. It should be noted that the imaging time and imaging quality of the BIM algorithm are affected by the choice of the degree of discretization and the number of iterations. In this paper, the parameters used uniformly were set as follows: the degree of discretization was set to (100, 100), and the number of iterations was set to 10 times.

Figure 9 shows the different locations and sizes of hemorrhagic strokes. From top to bottom, in order of scene from 0 to 7, a,d represents the ground truth, b,e are the result of BIM reconstruction, and c,f represents LEFEIM reconstructed dielectric constant distribution. BIM sucessfully reconstructed the stroke target of 10 mm, which were basically distinguishable and had very small positional errors. However, BIM ws less effective in reconstructing the different layers of each brain phantom. BIM was basically unable to distinguish the position of different layers and reconstruct their dielectric constant, and was barely able to reconstruct the shape of the relations between brain white matter and gray matter. It was difficult to distinguish the difference between stroke tissue and the gray matter in the case of a stroke with a radius of 5 mm. For LEFEIM, the resolution and accuracy of the reconstructed dielectric constant values were greatly improved compared to BIM. It was shown that LEFEIM can better restore the different layers of the brain, and the reconstruction was intuitively better than that of BIM. In terms of shape, the reconstruction of skin was slightly worse; however, the reconstruction of the cerebrospinal fluid (CSF) as well as the skull were better. The shape of gray matter and white matter was shown to be reconstructed, and hemorrhagic stroke could be distinguished. LEFEIM was also able to reconstruct the dielectric constant correctly. However, the shape reconstruction of stroke was relatively general, and could not produce clear circles. For strokes with a radius of 5 mm, the stroke was observed, but the calculation of the dielectric constant value was not as good as in scenarios with a larger stroke radius. For hemorrhagic strokes, both BIM and LEFEIM performed well, but LEFEIM still outperformed conventional BIM in the reconstruction of both resolution and shape.

Figure 10 shows ischemic strokes of different locations and sizes. In order from top to bottom and scenes 8–15, a,d represents ground truth, b,e is the result of BIM reconstruction, and c,f is the LEFEIM reconstructed dielectric constant distribution. For construction results deliverd by BIM, subtle differences can be observed in ischemic stroke reconstruction at different locations, but it is difficult to use this as a diagnostic basis. The approximate location could still be determined at a radius of 10 mm, but when the radius was 5 mm, it was difficult to determine whether the tissue was gray matter or ischemic stroke. The performance of BIM in reconstructing brain tissue between different layers was still poor, and it was almost unable to distinguish the difference in dielectric constant between different layers. For LEFEIM, its performance in distinguishing different brain tissues was significantly better than BIM, and the results of hemorrhagic stroke reconstruction were also better. However, it was still affected by reduced contrast, and the location and shape of ischemic strokes were not as accurate as hemorrhagic strokes. In terms of the accurate reconstruction of the dielectric constant, LEFEIM worked better in stroke reconstruction with a radius of 10 mm than with a radius of 5 mm. In scenes 8, 10, 12, and 14, a stroke radius of 10 mm could still be observed in terms of ischemic stroke. However, there was also the problem of poor shape reconstruction, and the reconstruction of the dielectric constant values for the ischemic type was not as good as for the hemorrhagic type. In the scenarios 9, 11, 13, 15, with a stroke radius of 5 mm, it was difficult to observe the reconstruction of the ischemic stroke. Overall, regardless of BIM or LEFEIM, the accuracy of dielectric constant reconstruction in ischemic stroke was generally lower than that in hemorrhagic stroke, but LEFEIM still outperformed BIM in distinguishing tissues and overall quality. There may be several reasons that lead to this problem. Firstly, the dielectric constant of ischemic stroke is close to that of gray matter, which greatly reduces contrast and affects reconstruction algorithms based on contrast. Secondly, LEFEIM did not make parameter adjustments or structural optimizations for data with close contrast, resulting in a decline in performance.

Figure 11 shows the test results of imaging using the experimental data, with specific stroke locations and radius referenced to scenarios 17 and 18 in Table 4. The proposed method still performed well in reconstructing the dielectric constant of hemorrhagic stroke. However, the results for ischemic stroke were still unsatisfactory, which is consistent with previous analysis.

In order to be able to quantitatively analyze the dielectric constant reconstruction of the brain, the results were quantified in three aspects. These included imaging time, root mean square error (RMSE), and coefficient of determination ($R^{2}$). RMSE as well as $R^{2}$ reflected the imaging performance of the different methods. The values in Table 5 reflect the analyzed data for BIM as well as LEFEIM, taken from the average of the test set. BIM (10) means the number of iterations was 10, and BIM (1) means the number of iterations was 1.

This subsection shows the brain imaging results of different methods in different scenarios. Compared to BIM, LEFEIM reduced imaging time, which may be due to BIM not using GPU acceleration, resulting in slower reconstruction speed. After extensive learning from the training set, LEFEIM was able to achieve faster and more accurate imaging. A larger RMSE indicates more outliers and poorer imaging, and the closer the coefficient of determination is to 1, the better the fitness of the regression model. From the perspective of RMSE and $R^{2}$, LEFEIM has better applicability and imaging performance in brain imaging.

In recent years, many new methods have been proposed, including Quadratic BIM [23] and AC-DBIM (Adaptive Clustering Distorted Born Iterative Method) [35]. To compare with these methods, $\zeta_{R}$ was introduced [36], which reflects the average error of all positions. The results are shown in Table 6. LEFEIM outperforms these new methods proposed in recent years in $\zeta_{R}$.

### 4.3. Noise Immunity Analysis

Noise is one of the unavoidable factors in practical applications. Thus, it is essential to evaluate a method’s ability to resist noise interference. In this subsection, in order to verify the noise immunity of the proposed model, the BIM iterative method is firstly compared with the proposed LEFEIM, and then the performance of the model is analyzed under different noise interference. Meanwhile, the scenarios used in this subsection are all related to the scenarios in Table 4. The adopted noise was Additive White Gaussian Noise (AWGN), where 10% noise indicates that the amplitude of the noise was 10% of the signal, and the phase of the noise was randomly assigned. Figure 12 shows the results of the brain reconstruction using BIM and the proposed model with 2% noise.

Figure 12 shows the comparison between BIM and LEFEIM imaging in a 2% noise environment, with the true dielectric constant distribution on the left, the BIM reconstruction in the center, and the LEFEIM imaging on the right. Although noise had a certain impact, the location of the stroke and the value of the dielectric constant can still be observed in the BIM reconstruction. However, it can be noticed that for the boundary reconstruction, it performed worse than the previous reconstruction, in which an elliptical shape in the brain can be observed. For LEFEIM, the location and a correct reconstruction of the dielectric constant values of the stroke can be seen. However, it can be clearly seen that the presence of noise causes artifacts in the imaging results, and the reconstruction of the boundaries was not as good as the previous measurement.

To further analyze the noise immunity of LEFEIM, the noise was incremented by a gradient of 2%. The experiment setup was consistent with the previous measurements. The reconstructed brain dielectric constant distribution is shown in Figure 13.

When the noise intensity was below 6%, the model still performed well and was able to distinguish the shape and boundary information of objects. The position of strokes could also be identified. However, when the noise intensity reached 10%. The reconstruction performance became degraded. Meanwhile, during the noise enhancement process, the background also produced some irregular information. This may have been due to the difficulty faced by the model in distinguishing between noise and useful signals, blurring the image.

The signal with added noise had a significant impact on the model, to some extent interfering with the reconstruction value of the dielectric constant, and the shape and position of stroke targets. This may be due to the fact that the training set of the model was not rich enough in data to have been trained with the added noise. However, even without targeted network structure design and training data, LEFEIM still showed some potential for noise immunity. At the same time, the performance in the low-noise case still had some credibility in being superior to BIM.

## 5. Conclusions and Future Work

This study built a brain tomography system and conducted imaging experiments using deep learning and BIM, and the proposed LEFEIM model achieved the expected results. The reconstructed dielectric constant distribution roughly reflected the real tissue distribution and could distinguish between stroke target and normal tissues. However, there were some limitations in the cases of ischemic strokes, added noise, and smaller strokes. Based on the analysis of the results of this experiment, LEFEIM can better perform the task of brain microwave tomography and is superior to BIM in terms of time and accuracy. One major advantage of the LEFEIM model is that the amount of data can be greatly enriched by the method of simulation, and combined with the actual data. On the one hand, this can alleviate the shortcomings of insufficient data for traditional model training and achieve better model training. On the other hand, it can improve the diversity of data types and enhance the generalization of the model. Second, both the computation time consumption and imaging performance of LEFEIM demonstrate an improvement over BIM to some extent. Third, this paper provides new ideas for brain microwave tomography by organically combining electric field computation, a microwave system, and deep learning. It is worth noting that in the application scenario of stroke imaging, microwave imaging has many inherent advantages, but microwave technology has not been used on a large scale in clinical practice due to time and imaging performance constraints. In response to these problems, LEFEIM represents a valuable attempt at a solution.

The plan for future work can be divided into two areas. On the one hand, at the time this research was carried out, the brain model used, as well as the signal and other parts of the brain, were actually simplified to a certain extent. In the future, we will develop scenes and a head phantom that are more reflective of reality. First of all, the scattered field calculation of transverse magnetic waves will be exchanged for a full-wave simulation, which means that the two-dimensional imaging will be further enhanced to three-dimensional imaging. At the same time, a new three-dimensional phantom needs to be created, with the spatial distribution of the brain tissue as close as possible to reality. Second, as many stroke locations as possible should be considered, and with reference to clinical experience, strokes should be categorized into three types: ischemic stroke, intracerebral hemorrhage, and subarachnoid hemorrhage, according to the location and the presence of hemorrhage or ischemia. Finally, after comprehensive and feasible experiments have been conducted, the production of an imaging system device can be considered for actual clinical testing and applications. On the other hand, in order to address the poor performance of measurement of ischemic stroke, which is essentially reflective of the problem of poor performance in low contrast situations, it is necessary to adjust the network structure in a targeted manner, increase the complexity of the model, and achieve better fitting.

## Figures and Tables

**Figure 1 sensors-24-06634-f001:**
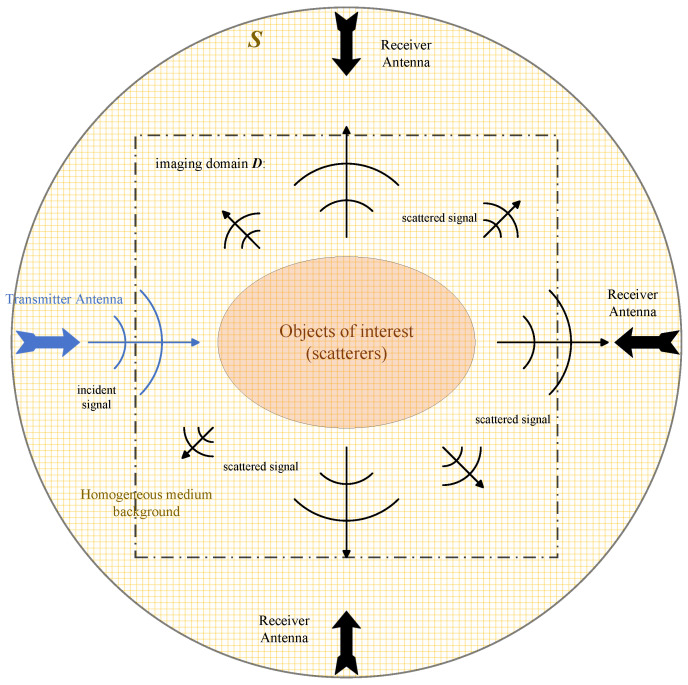
2D scattering scenario map.

**Figure 2 sensors-24-06634-f002:**
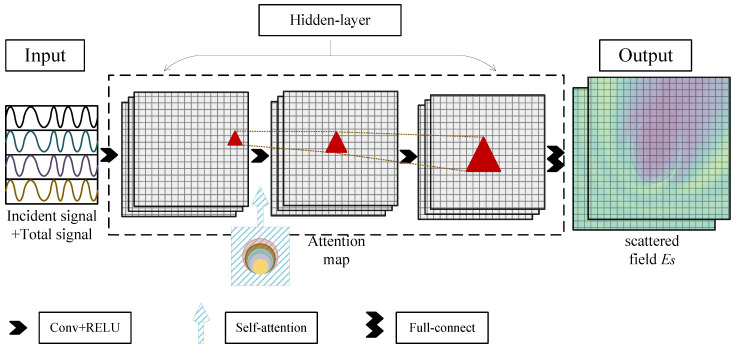
Schematic diagram of electric field enhancement learning network based on convolutional network.

**Figure 3 sensors-24-06634-f003:**
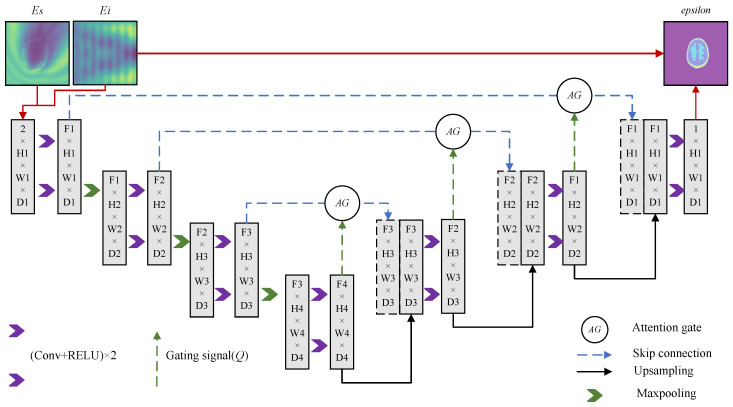
Schematic structure of AG U-net for dielectric constant reconstruction.

**Figure 4 sensors-24-06634-f004:**
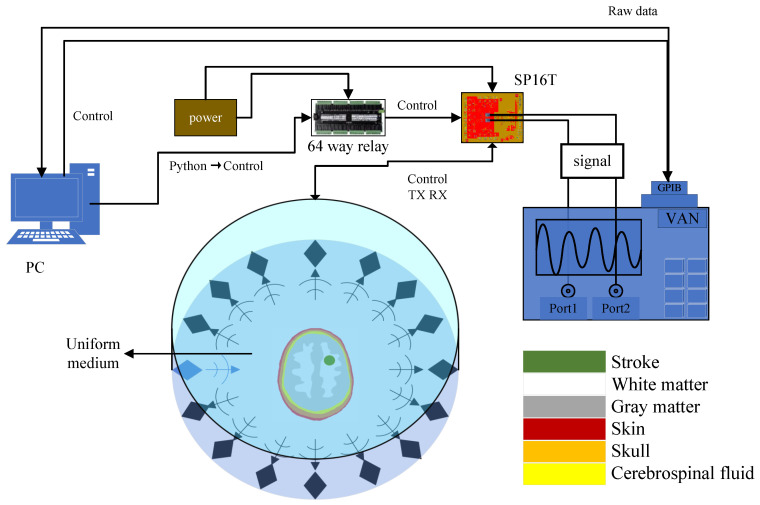
Imaging system.

**Figure 5 sensors-24-06634-f005:**
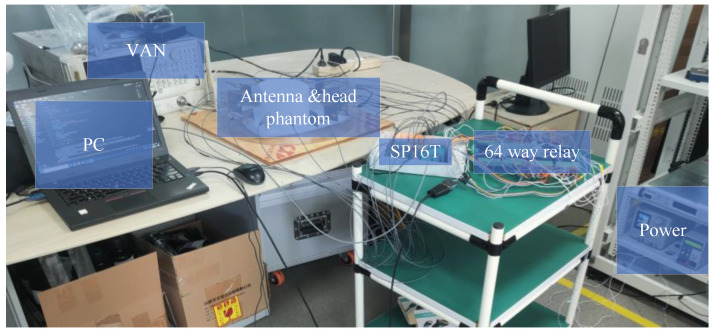
Overall view of the imaging system.

**Figure 6 sensors-24-06634-f006:**
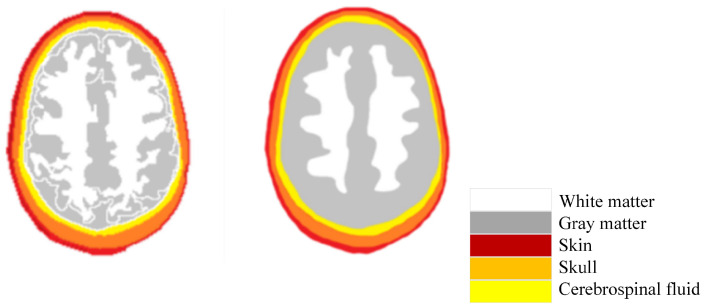
Slice #39 before and after modification.

**Figure 7 sensors-24-06634-f007:**
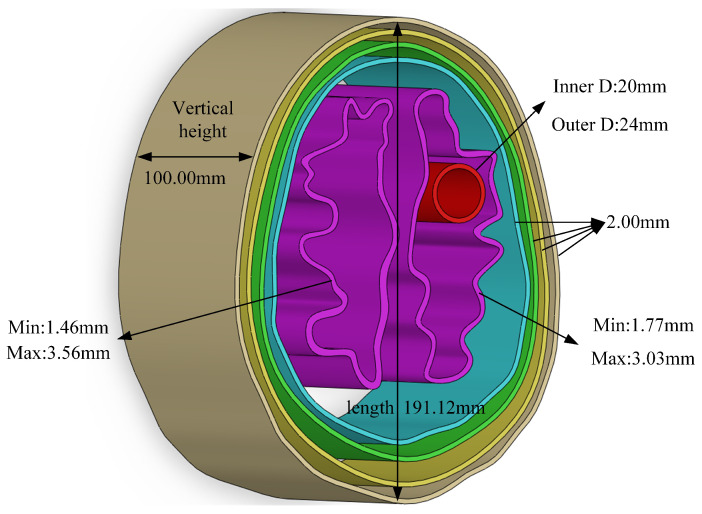
3D modeling based on slice #39.

**Figure 8 sensors-24-06634-f008:**
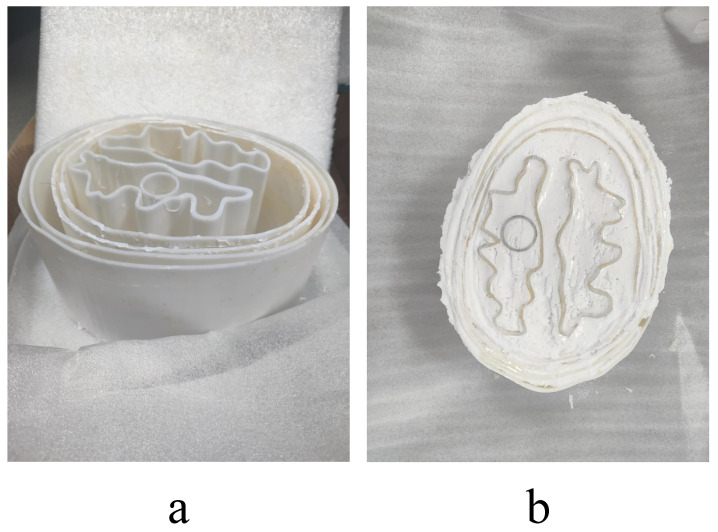
The Actual Production of the Head Phantom. (**a**) Top view. (**b**) Bottom surface.

**Figure 9 sensors-24-06634-f009:**
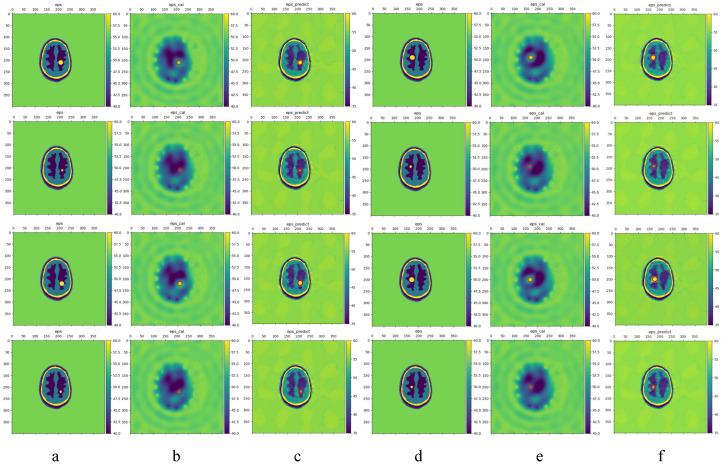
Comparison of the reconstructed dielectric constant distributions of eight hemorrhagic strokes. The red circle marks the true position. (**a**,**d**) Ground truth; (**b**,**e**) BIM; (**c**,**f**) LEFEIM.

**Figure 10 sensors-24-06634-f010:**
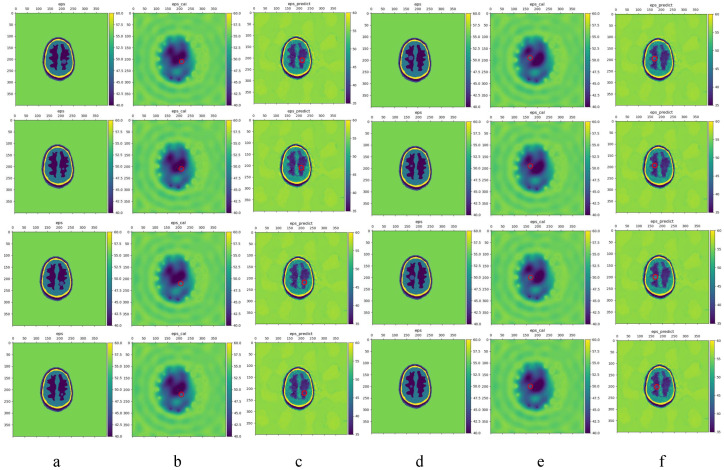
Comparison of the reconstructed dielectric constant distributions of eight ischemic strokes. The red circle marks the true position. (**a**,**d**) Ground truth; (**b**,**e**) BIM; (**c**,**f**) LEFEIM.

**Figure 11 sensors-24-06634-f011:**
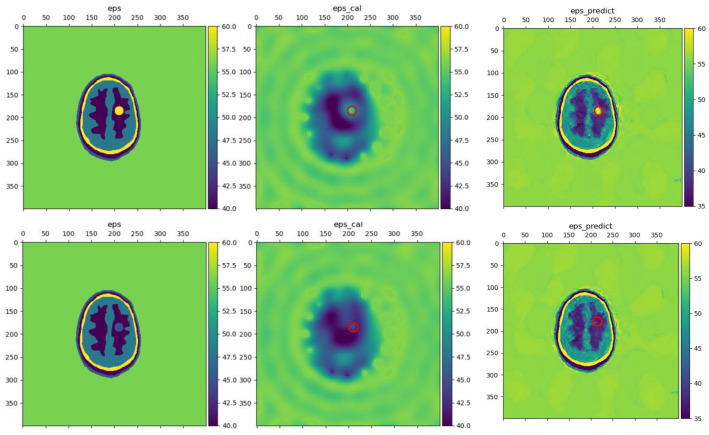
Reconstructed brain dielectric constant distribution using experimental data. The red circle marks the true position. Upper 1: hemorrhagic stroke ground truth; upper 2: BIM; upper 3: LEFEIM. Lower 1: ischemic stroke ground truth; lower 2: BIM; lower 3: LEFEIM.

**Figure 12 sensors-24-06634-f012:**
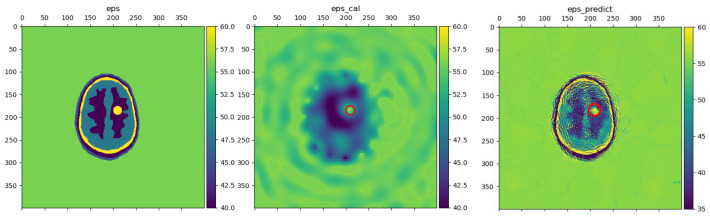
Imaging in 2% noise environments. The red circle marks the true position. Left: ground truth; center: BIM; right: LEFEIM.

**Figure 13 sensors-24-06634-f013:**
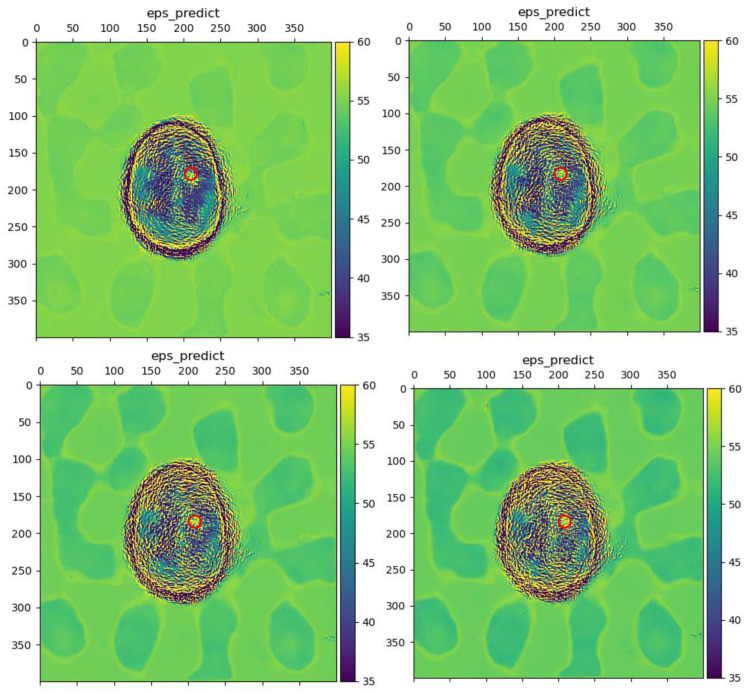
Reconstruction performance of the proposed LEFEIM under different intensities of noise. The red circle marks the true position. (**Top left**): reconstruction with noise intensity of 4%; (**top right**): reconstruction with noise intensity of 6%; (**bottom left**): reconstruction with noise intensity of 8%; (**bottom right**): reconstruction with noise intensity of 10%.

**Table 1 sensors-24-06634-t001:** Experimental equipment related parameters.

Device	Index
Height of phantom	100 mm
Number of antennas	16
Antenna Frequency	1.5 GHz
Antenna radius	250 mm
Imaging field edge length	400 mm, 400 mm
Stroke phantom radius	10 mm

**Table 2 sensors-24-06634-t002:** Parameters of the second-order Debye model for 100 MHZ–5 GHZ.

Brain Tissue	$\epsilon_{s}$	$\epsilon_{\infty }$	$\sigma_{s}$ (S/m)	τ (ps)
skin	38	4	0.0002	1856
skull	11	2.5	0.02	961
CSF	67	4	2	1212
Gray Matter	50	4	0.02	2269
White Matter	37	4	0.02	1850
ischemic stroke	30.00	/	0.5	/

**Table 3 sensors-24-06634-t003:** Head phantom equations for each tissue type.

Type	Volumes (mL)	Deionized Water (mL)	Gelatin Powder (g)	Kerosene Safflower Oil Mix (mL)
Skin	202	146	26	30
Skull	200	90	10	100
CSF	194	160	34	/
Gray matter	1080	630	90	360
White matter	1017	675	108	234
Ischemic stroke	30	16.5	3	10.5
Hemorrhagic stroke	30	24.9	5.1	/

**Table 4 sensors-24-06634-t004:** Setting for different stroke scenarios (using the center of the (400 mm, 400 mm) range as the zero point).

Scene Number	Stroke Type	Stroke Center Locations (mm)	Stroke Radius (mm)
0	hemorrhage	(10,10)	10
1	hemorrhage	(10,10)	5
2	hemorrhage	(20,10)	10
3	hemorrhage	(20,10)	5
4	hemorrhage	(−10,−30)	10
5	hemorrhage	(−10,−30)	5
6	hemorrhage	(0,−30)	10
7	hemorrhage	(0,−30)	5
8	ischemic	(10,10)	10
9	ischemic	(10,10)	5
10	ischemic	(20,10)	10
11	ischemic	(20,10)	5
12	ischemic	(−10,−30)	10
13	ischemic	(−10,−30)	5
14	ischemic	(0,−30)	10
15	ischemic	(0,−30)	5
16	health	/	/
17	ischemic	(−15,10)	10
18	hemorrhage	(−15,10)	10

**Table 5 sensors-24-06634-t005:** Comparison of the performance of LEFEIM with BIM.

Method	Cost Time (S)	RMSE	$R^{2}$
BIM (10)	209.73	27.18	0.41
BIM (1)	20.97	32.44	0.37
LEFE (GPU)	0.33	2.96	0.93

**Table 6 sensors-24-06634-t006:** Comparison of relative errors in dielectric constant of different methods.

Method	$\zeta_{R}$ [%]
LEFEIM	1.07
BIM	27.4
AC-DBIM (DUKE-1)	4.67
Quadratic BIM (Brain Slice 57)	48.1

## Data Availability

Data are contained within the article.

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
