# Peer review of "Deep Learning-Based Electric Field Enhancement Imaging Method for Brain Stroke"

_sensors, 2024, doi:10.3390/s24206634_

Round 1

Reviewer 1 Report

Comments and Suggestions for Authors

The paper presents several innovative contributions in applying deep learning to the medical imaging domain, improving the speed and accuracy of stroke detection using microwave tomography. There are several parts can be improved in the paper.

There is no clear rationale given for the specific choices in the model architecture of CNN (layer number and kernel size).

The explanation of the phantom construction, while informative, is somewhat long and could be condensed.

Authors should discuss more why ischemic stroke reconstruction is less effective and provide possible future improvements.

The references at Line 72 is missing.

Comments on the Quality of English Language

The abbreviation is not explained when first introduced.

The manuscript contains some phrasing, grammatical, and stylistic issues that could be improved. For instance, lines 196 to 201 are repeated. And phrases like 'It can be seen' are unnecessary. The paper also occasionally shifts between past and present tense, which should be made consistent throughout.

Author Response

Comments 1: There is no clear rationale given for the specific choices in the model architecture of CNN (layer number and kernel size).

Response1:  We thank the reviewers for providing constructive feedback. We have fully revised ourmanuscript and have addressed all of the reviewers' comments, as well as added newanalyses to further strengthen our work.  The reasons for CNN network design are added to line 161 of the article.  In fact, the electric field resolution enhanced by CNN is 400 * 400.  Stacking multiple 3 * 3 small convolution kernels can have the same receptive field as large convolution kernels,  but the parameters and computational complexity of small convolution kernels are less.  Stacking multiple 3 * 3 small convolution kernels can introduce more nonlinearity compared to large convolution kernels,  resulting in better reconstruction of the scattering field.

Comments 2: The explanation of the phantom construction, while informative, is somewhat long and could be condensed.

Response2:  Thank you for pointing this out.  We agree with this comment.  Due to the interdisciplinary nature of the research direction,  which involves the dielectric constant of tissues,  electromagnetic scattering theory,  and deep learning,  it is elaborated in detail to express it more clearly.  We will streamline the content of this section in the future.

Comments 3:Authors should discuss more why ischemic stroke reconstruction is less effective and provide possible future improvements.

Response3:  We thank the reviewers for providing constructive feedback.  We conducted a more detailed analysis at 460-481,  including why this phenomenon occurs and methods that can be attempted in subsequent papers.  Added possible attempts to address this issue in line 569.

Comments 4:The references at Line 72 is missing.

Response4:  Thank you for pointing this out.  We have corrected the 72 missing references.

We thank the reviewers for providing constructive feedback.  We will continue to improve the errors in English expression.

Reviewer 2 Report

Comments and Suggestions for Authors

The article “Deep Learning-based Electric Field Enhancement Imaging Method for Brain Stroke” is devoted to stroke research. Machine learning methods are used to analyze microwave tomography system data, and two approaches to analyzing these data are proposed. The article is interesting, but there are several questions:

1. In the text of the article is given the architecture of the model in Fig. 2, which uses convolutional layers of the neural network, is it fashionable to use instead of these layers methods of dimensionality reduction such as UMAP or Isomap?

2. Why U-Net standard metrics were not applied?

There are many punctuation errors in the text of the article

Author Response

Comments 1: In the text of the article is given the architecture of the model in Fig. 2, which uses convolutional layers of the neural network, is it fashionable to use instead of these layers methods of dimensionality reduction such as UMAP or Isomap?

Response1: Thank you to the reviewer for raising this question.  Regarding this issue,  according to our research,  methods UMAP and Isomap are not particularly popular in the field of microwave imaging combined with deep learning.  When considering dimensionality reduction algorithms,  UMAP and Isomap have not been taken into account temporarily due to concerns about high computational complexity and expensive computational costs.  Meanwhile,  due to the need for parameter optimization in UMAP,  it has not been used temporarily. But these will be attempted and analyzed in subsequent work.

Comments 2:Why U-Net standard metrics were not applied?

Response2:  Thank you for pointing this out.  In fact,  in this study,  CNN and U-Net was evaluated as a whole,  corresponding to BIM (i.e. traditional electromagnetic backscatter method).  So this study essentially fits the proposed method for electromagnetic backscatter problems,  considering the coefficient of determination as a regression task.  At the same time,  root mean square error was added to evaluate the imaging quality of the overall method,  and the time advantage was compared.  So U-Net's standard metrics were not used.

Comments 3:There are many punctuation errors in the text of the article

Response3: Thank you for pointing this out.  We agree with this comment.  We have made corrections.

Reviewer 3 Report

Comments and Suggestions for Authors

Brain imaging methods such as CT, MRI, and PET, which are standard in clinical practice but have limitations in stroke diagnosis due to high costs and lack of portability. Microwave imaging, as an emerging technique, offers a more cost-effective and portable solution, requiring computational algorithms for enhanced processing speed and resolution. In the paper titled "Deep Learning-based Electric Field Enhancement Imaging Method for Brain Stroke", the authors introduce "LFEFEIM", a two-step deep learning framework named BIM to address this challenge.

However, I have the following concerns:

1. The authors only compared their proposed method with BIM, a technique developed 30 years ago. There have been several recent advancements in this area within the past two years, which should be acknowledged. It is crucial for the authors to outline the development of BIM, as well as U-net, in the introduction and include a comparison with these recent methods in the evaluation section.

- Costanzo, S., Flores, A., & Buonanno, G. (2023, October). Microwave Imaging for Brain Cancer Detection: Enhanced Accuracy with Machine Learning Approach. In 2023 IEEE International Conference on Metrology for eXtended Reality, Artificial Intelligence and Neural Engineering (MetroXRAINE) (pp. 519-524). IEEE.

- Yao, H. M., Zhang, H. H., Jiang, L., & Ng, M. (2024). Fast Electromagnetic Inversion Solver Based on Conditional Generative Adversarial Network for High-Contrast and Heterogeneous Scatterers. IEEE Transactions on Antennas and Propagation.

- Yao, H. M., Jiang, L., & Ng, M. (2024). Enhanced Deep Learning Approach Based on the Conditional Generative Adversarial Network for Electromagnetic Inverse Scattering Problems. IEEE Transactions on Antennas and Propagation.

- Zhang, H. H., Yao, H. M., Jiang, L., & Ng, M. (2022). Enhanced two-step deep-learning approach for electromagnetic-inverse-scattering problems: Frequency extrapolation and scatterer reconstruction. IEEE Transactions on Antennas and Propagation, 71(2), 1662-1672.

- Wang, Y., Zhao, Y., Wu, L., Yin, X., Zhou, H., Hu, J., & Nie, Z. (2023). An Early Fusion Deep Learning Framework for Solving Electromagnetic Inverse Scattering Problems. IEEE Transactions on Geoscience and Remote Sensing, 61, 1-14.

2. There are missing references in the first paragraph of the introduction, which need to be included for completeness.

Furthermore, there are several issues with the writing style:

1. The sentence "This section mainly reviews previous studies" (line 34) is redundant and should be removed.

2. Placeholder "[??]" remains unaddressed in line 72.

3. There are instances of duplicate and missing spaces that need correction.

These revisions will improve the clarity and academic rigor of the paper.

Author Response

Comments 1: The authors only compared their proposed method with BIM, a technique developed 30 years ago. There have been several recent advancements in this area within the past two years, which should be acknowledged. It is crucial for the authors to outline the development of BIM, as well as U-net, in the introduction and include a comparison with these recent methods in the evaluation section.

Response1:  We agree with the commentator's viewpoint that more discussion is needed on the methods of recent years.  We conducted research and analysis on some new methods in the past two years in line 75 of the introduction,  and added recommended references to the reference list.

Comments 2: There are missing references in the first paragraph of the introduction, which need to be included for completeness.

Response2:  Thank you for pointing out the issues.  We have added the supplementary references to lines 643-656.

Comments 3: Furthermore, there are several issues with the writing style:1. The sentence "This section mainly reviews previous studies" (line 34) is redundant and should be removed.2. Placeholder "[??]" remains unaddressed in line 72.3. There are instances of duplicate and missing spaces that need correction.

Response3:  We thank the reviewers for providing constructive feedback.  We agree with this comment. The sentence in line 34 has been revised, and the missing reference in line 72 has also been supplemented.  Other language issues have also been corrected.

Round 2

Reviewer 3 Report

Comments and Suggestions for Authors

The authors address all my concerns except for missing comparison with recent methods in the evaluation section.

Author Response

Comments 1: The authors address all my concerns except for missing comparison with recent methods in the evaluation section.

Response1:  We thank the reviewers for providing constructive feedback. To compare with new methods in recent years, we calculated the new metric ζR values of LEFEIM and BIM and added a table located from line 505 to 510.

Round 3

Reviewer 3 Report

Comments and Suggestions for Authors

The authors address all my concerns.